Review  

**Subject Area:**
genetics/genomics/developmental biology/neuroscience/cellular biology

X chromosome inactivation, chromatin, long non-coding RNA, epigenetic

**Author for correspondence:**
Neil Brockdorff
e-mail: neil.brockdorff@bioch.ox.ac.uk

# Localized accumulation of Xist RNA in X chromosome inactivation

Neil Brockdorff

Department of Biochemistry, University of Oxford, South Parks Road, Oxford OX1 3QU, UK

NB, 0000-0003-4838-2653

The non-coding RNA Xist regulates the process of X chromosome inactivation, in which one of the two X chromosomes present in cells of early female mammalian embryos is selectively and coordinately shut down. Remarkably Xist RNA functions *in cis*, affecting only the chromosome from which it is transcribed. This feature is attributable to the unique propensity of Xist RNA to accumulate over the territory of the chromosome on which it is synthesized, contrasting with the majority of RNAs that are rapidly exported out of the cell nucleus. In this review I provide an overview of the progress that has been made towards understanding localized accumulation of Xist RNA, drawing attention to evidence that some other non-coding RNAs probably function in a highly analogous manner. I describe a simple model for localized accumulation of Xist RNA and discuss key unresolved questions that need to be addressed in future studies.

## 1. Introduction

X chromosome inactivation is the process that evolved in mammals to equalize levels of X-linked gene expression in XX females relative to XY males. In early female embryos genes on both X chromosomes are expressed at normal levels. As cell differentiation and development proceed, each cell randomly selects and coordinately silences one of the two X chromosomes [1]. Chromosome silencing occurs as a result of modifications to the structure and organization of the underlying chromatin, with specific changes in post-translational modifications of histone tails, incorporation of variant histones, DNA methylation of promoters of X-linked genes and remodelling of the higher order structure of the chromosome [2]. The heterochromatic state of the inactive X chromosome (Xi), once established, is highly stable, and is transmitted epigenetically through successive cell generations during development and into adult life.

Studies in the 1990s proposed a long non-coding (lnc) RNA, conserved in human and mouse, and transcribed specifically from the Xi, as having a central role in regulating the X inactivation process [3–7]. Intriguingly, this lncRNA, denoted as the X inactive specific transcript (XIST/Xist in human and mouse respectively; mouse nomenclature is used throughout this review), was found to accumulate locally *in cis*, circumscribing a discrete nuclear subdomain that overlaps with Xi chromatin in the interphase cell nucleus [8]. The behaviour of Xist RNA is distinct from that of other mRNAs for which only unprocessed nascent transcripts exhibit local accumulation at the site of synthesis [9]. Molecular analyses determined that the Xist RNA is a 15–17 kb RNA polymerase II (RNAPII) transcript that is both spliced and polyadenylated [4,7]. Approximately 40% of processed Xist sequence comprises tandem repeat elements, designated A–F, each unique with respect to one another and, in large part, with respect to other sequences in the genome. Primary sequence homology between mouse and human sequences varies in different regions, ranging from high to moderate/low, with several of the tandem repeat elements A–F being highly conserved [4,7,10].

Gene knockout and transgenic experiments in mouse demonstrated that Xist is both necessary and sufficient for X inactivation [11,12]. Notably, Xist transgenes

royalsocietypublishing.org/journal/rsob   Open Biol. 9: 190213

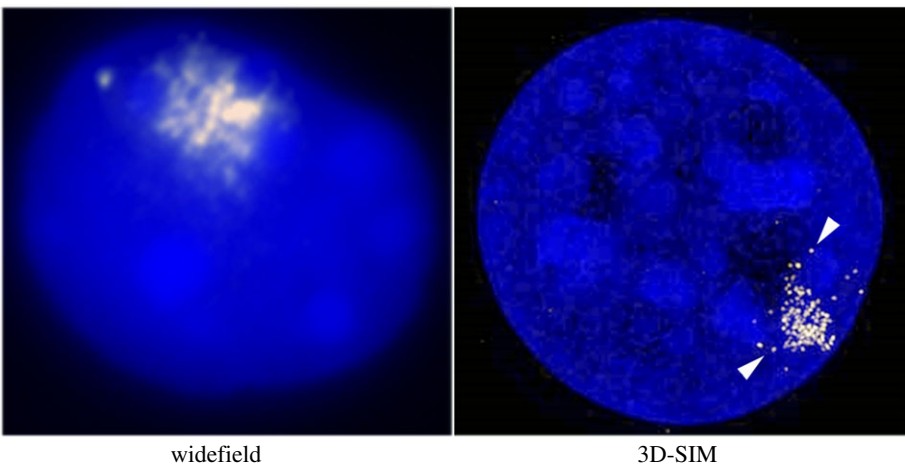

| widefield | 3D-SIM |

**Figure 1.** Xist RNA clouds detected using RNA-FISH. The examples shown each illustrate a single XX cell nucleus at interphase. The Xist RNA-FISH signal is shown in yellow and the DNA counterstain, DAPI, in blue. Using conventional widefield microscopy (left) Xist RNA signal appears as a cluster of large foci occupying a discrete nuclear domain. Super-resolution 3D-SIM (right) resolves the Xist RNA signal as punctate foci thought to represent single Xist RNA molecules (arrowheads).

located on autosomes were found to behave similarly to the Xist gene on the X chromosome, giving rise to both localized accumulation of Xist RNA *in cis* and chromosome silencing of the chromosome bearing the Xist transgene [12].

With the discovery of Xist RNA in the early 1990s, the field turned to three important questions: (i) How is the Xist gene regulated so as to ensure that Xist expression occurs only on a single X chromosome in female cells? (ii) How does Xist RNA accumulate locally on the chromosome from which it is transcribed? (iii) How does Xist RNA bring about formation of repressive chromatin and gene silencing? In this perspective I focus on our progress towards understanding the second question, the molecular basis for localized accumulation of Xist RNA. Readers interested in advances in understanding Xist gene regulation and Xist-mediated chromatin modification are referred to other recently published reviews [13,14].

Understanding localized accumulation of Xist RNA presents several key challenges: How does Xist RNA evade transport from the nucleus to the cytoplasm? How does Xist RNA spread from the site of synthesis? Does Xist RNA spreading involve passive or active processes? Does Xist RNA have preferred binding sites? How is Xist RNA accumulation limited to a single chromosome territory? Although we currently have few definitive answers to these questions several interesting clues have begun to emerge. I will summarize this recent progress, setting out competing models and hypotheses, and will additionally provide suggestions for how these may be addressed experimentally in future studies.

## 2. Xist RNA accumulates *in cis*

Classical genetic studies led to the proposal that there is a *cis*-acting locus on the X chromosome, referred to as the X inactivation centre (XIC), which must be present for that chromosome to undergo X inactivation. It was through efforts to pinpoint the XIC that Xist was discovered [3,5,6,15–17]. The fact that Xist RNA is expressed exclusively from the Xi was suggestive of a link to action *in cis*. This possibility was reinforced by cell fractionation experiments showing that Xist RNA is retained in the cell nucleus [7] and RNA fluorescence *in situ* hybridization (FISH) analyses that demonstrated Xist RNA has the appearance of a discrete

nuclear domain comprising clustered foci overlying the dense inactive X chromatin (Barr body) [4,8]. This pattern of Xist RNA localization is often referred to as a 'cloud'. More recently, super-resolution imaging has been applied to further resolve the clustered foci within Xist RNA clouds, revealing individual Xist RNA molecules [18–20]. Representative examples illustrating localized accumulation of Xist RNA detected by RNA fluorescence *in situ* hybridization (RNA-FISH), using either conventional widefield or super-resolution three-dimensional structured illumination microscopy (3D-SIM) are shown in figure 1. While the majority of Xist RNA detected by RNA-FISH localize within a discrete domain, it cannot be ruled out that a proportion of transcripts translocate beyond the confines of the single chromosome territory. Ascertaining whether or not this is the case is challenging as DNA-FISH methods used to label a single chromosome territory in combination with Xist RNA-FISH require relatively disruptive conditions and, moreover, fail to reveal all of the underlying chromosome sequences, for example dispersed repeat sequences.

Localized accumulation of Xist RNA over a single chromosome territory was substantiated by Duthie *et al.* [21], who observed that Xist RNA decorates one of the two X chromosomes at metaphase in rodent cell lines. This study further showed that Xist RNA dissociates from the chromosome at anaphase/telophase, with Xi-specific accumulation being re-established during early G1. This finding contrasts with analysis of human cell lines in which Xist RNA dissociates from Xi at prophase [8]. In later work it was reported that chromatin changes mediated by the mitotic kinase Aurora B modulate binding versus release of XIST RNA from the mitotic Xi chromosome [22].

Detailed analysis of Xist RNA on Xi in mitotic cells revealed that Xist transcripts are concentrated over gene-rich chromosome domains and excluded from gene-poor domains, including pericentromeric heterochromatin, resulting in a banded appearance [21]. This observation indicates either that Xist RNA localizes to preferred sites in gene-rich chromosome domains or that Xist RNA is preferentially lost from gene-poor domains as cells enter mitosis. The former hypothesis is supported by immunofluorescence, and to some degree by high-throughput sequencing (HTS) analysis of Xist-mediated chromatin modifications [23–25]. Conversely it

has been observed that the banded appearance of Xist RNA on mitotic Xi chromosomes becomes more pronounced as cells progress towards anaphase [26]. Additionally, two recently developed methods, RAP-seq [27] and CHART-seq [28], which map Xist RNA occupancy at base pair resolution, indicate that Xist RNA associates to a greater or lesser extent over the entire Xi chromosome. It should however be noted that gene-poor domains may be significantly underrepresented in short-read HTS-based methods because highly repetitive sequences are often unmappable.

Since the discovery of Xist RNA other examples of regulatory RNAs proposed to mediate their effects via localized accumulation *in cis* have emerged. The lncRNA Rsx, which is evolutionarily unrelated to Xist, performs an equivalent function in X inactivation in marsupial mammals, accumulating over the Xi chromosome *in cis* and silencing underlying genes by inducing chromatin modification [29]. Similar to Xist RNA, the Rsx transcript is large and includes several blocks of tandemly repeated sequences. This is likely to be a key feature of lncRNAs in relation to evolving localized accumulation and silencing functions, as discussed previously [30]. It is currently unknown if the RNA-binding proteins (RBPs) implicated in Xist RNA function also play a role in Rsx-mediated X inactivation. Nevertheless, this example serves to illustrate an independent case where localized accumulation of lncRNAs *in cis* has evolved to coordinately repress underlying genes. Other lncRNAs thought to function similarly to Xist include Kcnq1ot1 and Airn, which are required for imprinted repression of large chromosomal regions on mouse chromosomes 7 and 17, respectively [31–33]. In both cases there is evidence for localized accumulation of the lncRNA *in cis* with the site of transcription [34,35]. Further biochemical and mechanistic studies are required to fully explore the similarities and differences between Rsx, Kcnq1ot1, Airn and Xist RNA.

While there is an overall consensus that Xist RNA accumulates *in cis*, Jeon & Lee [36] reported *trans*-localization of Xist RNA from the Xi chromosome to autosomes that bear an inducible Xist RNA transgene. Specifically, integration of the inducible Xist transgene in female mouse embryo fibroblasts (MEFs) resulted in reduced Xist RNA clouds on an endogenous Xi chromosome (referred to as squelching), and, following transgene induction, Xist RNA from the Xi was detected in association with the transgene-expressing chromosome, indicating that Xist transcripts can also localize to sites *in trans*. The latter finding was based on FISH analysis using a probe specific for Xist RNA produced on the endogenous Xi. Further analysis identified a short region of Xist RNA centred on repeat F, which binds the nuclear protein YY1 and which is required for transgenic Xist RNA accumulation and squelching. However, subsequent studies have found that the repeat F region is the major enhancer required for Xist RNA expression [37]; in light of this, the squelching effect could be attributable to competition for positive-acting transcription factors required for Xist expression. Consistent with this idea, more recent studies have reported that deletion of the repeat F region does not impact localized accumulation of Xist transgenes driven from a heterologous inducible promoter [38,39].

The accumulation of endogenous Xist RNA *in trans* at expressed Xist transgene sites, as reported by Jeon & Lee [36], is difficult to reconcile with the majority of studies analysing the behaviour of Xist RNA. Accordingly, further investigations are required to confirm these observations and to rule out possible technical artefacts. In this context a plausible explanation for the detection of *trans*-located Xist transcripts at Xist transgene sites is low-level plasmid DNA contamination of the RNA-FISH probes. This is because Xist transgenes generally co-integrate with, and therefore potentially transcribe, associated plasmid sequences. Taking this uncertainty into consideration, in the context of this review I will focus on the generally held consensus that localized accumulation of Xist RNA over a defined chromosome territory occurs strictly *in cis*.

## 3. Xist RNA is anchored through interactions with proteins of the nuclear matrix

In an important early study Clemson *et al.* [8] reported that Xist RNA clouds, as detected by RNA-FISH, are tightly bound up in the nuclear matrix. Classically, the term nuclear matrix refers to a reticular network of proteins and RNA that is retained in cell preparations following digestion of DNA and extensive salt extraction [40]. Studies over several years defined that the major constituents of this biochemical fraction comprise structural proteins, chaperones, DNA/RNA-binding proteins, chromatin remodelling and transcription factors and RNA. The functional significance of the nuclear matrix has been the subject of considerable debate over many years, with it having suggested roles in providing a super-structure for key nuclear processes such as DNA replication, through to the proposal that the nuclear matrix is an artefact of the biochemical procedures by which it is isolated. Regardless, Xist RNA is clearly bound up among the proteins present in matrix-extracted nuclei, apparently independent of interactions with DNA/chromatin. One consequence of this finding is that efforts to purify native Xist RNA/protein complexes from cells have been severely hampered, a limitation that has yet to be overcome.

A further link between localized accumulation of Xist RNA and the nuclear matrix came through the discovery that the major nuclear matrix protein scaffold attachment factor A (SAF-A)/heterogeneous nuclear ribonucleoprotein U (hnRNPU) has, has a role in the formation of Xist RNA clouds [41]. In this study the authors performed a loss of function screen (RNAi knockdown), testing the requirement for several known RBPs in Xist RNA accumulation. Knockdown of SAF-A/hnRNPU, but not other RBPs, resulted in dispersal of Xist RNA throughout the nucleoplasm. Similar findings were subsequently reported for different rodent cell lines [42,43], although analysis of transformed human cell lines indicates that other factors may function redundantly with SAF-A/hnRNPU [44]. Consistent with this idea, hnRNPUL1, a direct homologue of SAF-A/hnRNPU, is able to partially restore Xist cloud formation following SAF-A/hnRNPU knockdown in rodent cells [45]. Other studies have reported that SAF-A/hnRNPU is concentrated over the Xi territory at interphase [44,46], although there appears not to be a simple linear relationship with the onset of Xist RNA expression [46].

SAF-A/hnRNPU has been reported to bind Xist RNA directly, and this was linked to the presence of an RGG domain at the C-terminal end of the protein [41,47]. Whether or not this interaction is specific is open to debate as UV cross-

royalsocietypublishing.org/journal/rsob    Open Biol. 9: 190213

linking experiments indicate binding throughout Xist RNA [41,48], a finding substantiated in transcriptome-wide mapping of SAF-A/hnRNPU-binding sites using (UV) cross-linking immunoprecipitation (CLiP) assays [49]. In more recent work SAF-A/hnRNPU was found to undergo ATP-dependent polymerization in the presence of RNA, triggered in cells by chromatin-associated RNAs [50], again indicating a relatively non-specific interaction of the protein with different RNAs. In this study SAF-A/hnRNPU polymerization was suggested to have a role in decompacting chromatin [50], running counter to what might be expected in the context of X inactivation, so the role of this protein clearly requires further investigation.

Further evidence for co-association of Xist RNA and SAF-A/hnRNPU comes from an analysis using 3D-SIM super-resolution microscopy which found that Xist RNA co-localizes with SAF-A/hnRNPU within interchromatin spaces, more specifically at perichromatin boundaries [20]. Notably markers of Xi chromatin did not co-localize with Xist RNA at this resolution, indicating physical separation of Xist RNA and the chromatin on which it acts.

More recently, a second nuclear matrix-associated factor, CDKN1A interacting zinc finger protein 1 (CIZ1), was reported to have a role in localized accumulation of Xist RNA [51,52]. CIZ1 co-localizes with Xist RNA and moreover binds Xist RNA within a defined sequence element, the E-repeat. The basis for this interaction has not been fully elucidated but is likely to involve an RNA-binding zinc finger domain located towards the C-terminal end of the protein. Similar to SAF-A/hnRNPU, CIZ1 loss of function in fibroblasts or lymphocytes results in dispersal of Xist RNA throughout the nucleoplasm [51]. Strikingly, restoration of CIZ1 in knockout MEFs leads to re-establishment of normal Xist RNA domains, indicating that localized accumulation is a direct consequence of CIZ1 binding to Xist RNA.

The essential requirement for CIZ1 in localized accumulation of Xist RNA is thought to be limited to somatic cell types. Specifically, CIZ1 null female mice are viable [51], indicating that there is no defect in localized Xist RNA accumulation during early development, when X inactivation is established. Additional evidence that CIZ1 function is required only in somatic cells comes from an independent study that reported a role for Xist exon 7, which encompasses the E-repeat required for CIZ1 binding, in localized accumulation of Xist RNA [48]. Here, an Xist exon 7 truncated allele expressed in differentiating XX embryonic stem (ES) cells was seen to form Xist clouds with reduced efficiency, but only at later differentiation time points. It is interesting to note that, although CIZ1-dependent Xist localization is apparently limited to somatic cell types, CIZ1 protein co-localizes with Xist RNA clouds in all cell types and at all developmental stages. This observation indicates that there are compensating factors that function redundantly with CIZ1 in early embryonic cell lineages.

While CIZ1 null mice are viable, indicating that there is no major deficit in X inactivation, ageing animals do exhibit a female-specific lymphoproliferative disorder. A likely explanation for this finding is that CIZ1 has a critical role in localizing Xist RNA in activated lymphocytes. Specifically, resting female T- and B-cells from wild-type animals extinguish Xist RNA expression and then restore Xist clouds in response to antigen stimulation [53]. Following antigen stimulation of resting T- and B- cells from CIZ1 null animals, Xist RNA exhibits a highly dispersed localization [51]. Thus, the supposition is that aberrant Xist RNA re-localization results in deficiencies

in X inactivation in activated T- and B-cells, and that this in turn leads to uncontrolled cell proliferation of lymphocytes.

The inter-relationship between CIZ1 and SAF-A/hnRNPU in localized accumulation of Xist RNA is not yet understood. In one study it was reported that SAF-A/hnRNPU knockdown in MEFs phenocopies loss of CIZ1 (Xist RNA dispersal), and moreover that CIZ1 binding to the dispersed Xist RNA molecules is retained [52]. The simplest explanation for this observation is that CIZ1 and SAF-A/hnRNPU function in the same pathway for localizing Xist RNA transcripts. One possibility, illustrated in figure 2a, is that SAF-A/hnRNPU undergoes a local ATP-dependent polymerization upon binding to Xist RNA, and that this provides a reticular substrate for anchoring Xist RNA molecules via interactions with CIZ1. To account for cell type-specific effects of CIZ1 we can speculate, as above, that other sequence-specific Xist RBPs can also function as anchoring factors, acting redundantly with CIZ1, notably in early embryogenesis when X inactivation is first established (figure 2b). This model is consistent with evidence indicating redundancy in the mechanism of Xist localization, at the level of both binding sites [54] and, as above, Xist interacting factors.

# 4. On the nature of Xist RNA-anchoring sites

Although it is clear that nuclear matrix proteins are important for anchoring Xist RNA, there remain many questions in relation to the nature of anchoring sites. An interesting observation is that the abundance of Xist RNA in dividing cells increases approximately twofold at S-phase [55], suggesting that each sister chromatid has a defined capacity for anchored Xist RNA particles. This finding raises the question of whether there are preferred sites to which Xist RNA anchors or whether the whole chromosome is available, with only a limited number of sites being open at a given time? Related to this point, anchoring sites may be defined by their affinity for Xist RNA particles or, alternatively, by their accessibility for Xist RNA binding.

What then can we deduce about the nature of anchoring sites? As mentioned above, Xist RNA is concentrated over gene-rich domains and excluded from gene-poor regions and constitutive heterochromatin, indicating a link between active chromatin and Xist RNA anchoring. Interestingly, SAF-A/hnRNPU polymerization requires active transcription [50], and this could be relevant in terms of defining anchoring sites at the onset of X inactivation. Anchoring does not appear to involve a base-pairing interaction with underlying DNA, as evidenced by resistance of Xist RNA clouds to RNAseH treatment, which digests DNA/RNA hybrids [8] and by 3D-SIM super-resolution microscopy indicating significant spatial separation of Xist RNA and chromatin [18,20]. Additionally, the fact that autosomally integrated Xist transgenes display localized accumulation in cis demonstrates that anchoring sites are not unique to the X chromosome.

More recent progress has come from studies that applied RAP-seq [27] and CHART-seq [28] to map Xist RNA occupancy in cell populations at base-pair resolution. Briefly, short probes designed to complement Xist RNA sequence were used to affinity purify formaldehyde cross-linked sonicated chromatin fragments from cell populations that are expressing Xist RNA. HTS of the associated chromatin generates high-resolution

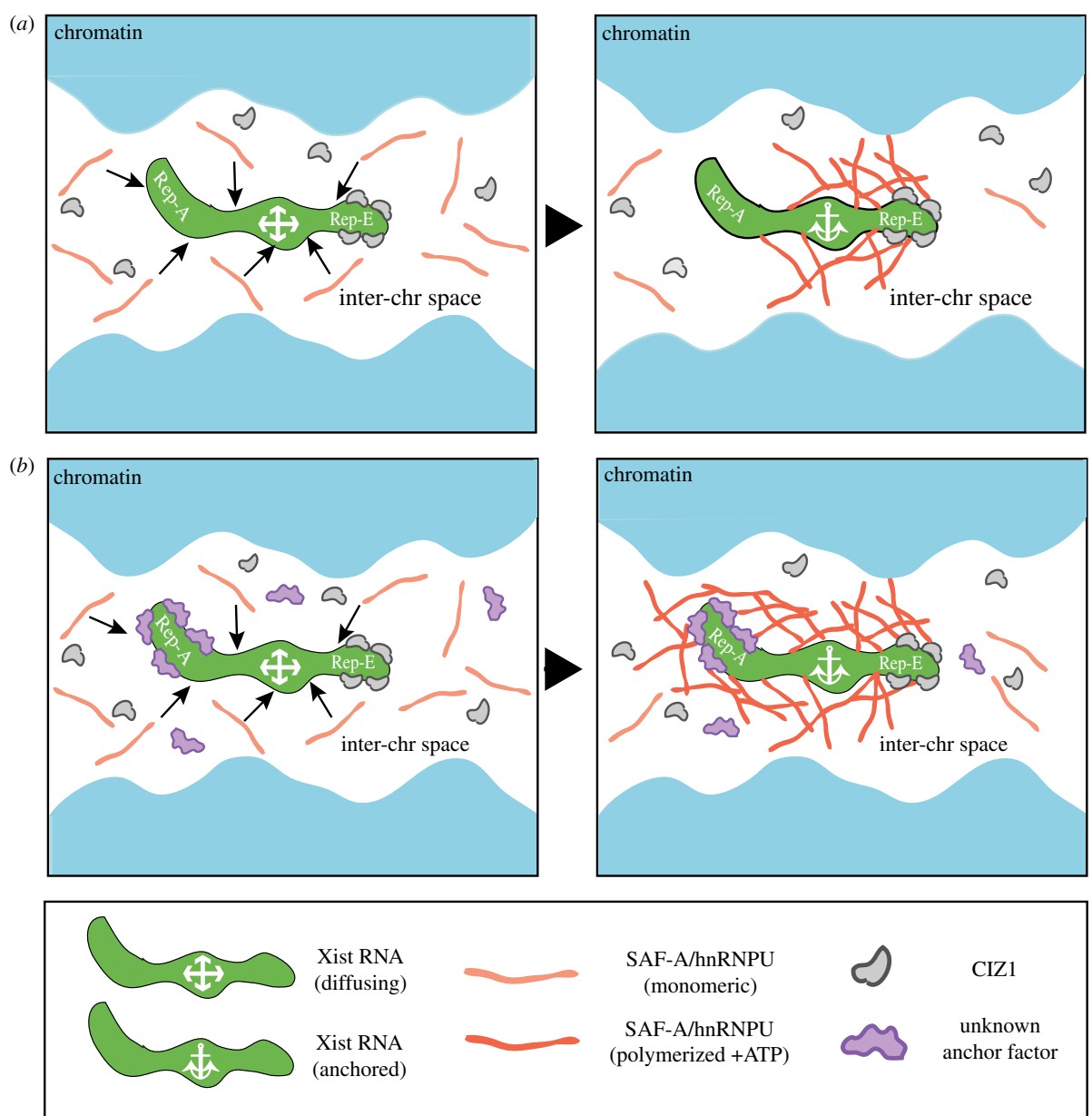

**Figure 2.** Proposed model for the interplay between SAF-A/hnRNPU and CIZ1 (*a*) and/or alternative anchoring factors (*b*) in localized accumulation of Xist RNA. (*a*) SAF-A/hnRNPU monomers bind to Xist RNA molecules that are diffusing within inter-chromatin (inter-chr) spaces (left panel, arrows), triggering ATP-dependent SAF-A/hnRNPU polymerization (right panel). CIZ1 bound to the Xist E-repeat (Rep-E) anchors the Xist molecule through an interaction with the polymerized SAF-A/hnRNPU scaffold. (*b*) Other anchoring factors not present in lymphocytes or fibroblasts function redundantly with CIZ1. Illustrated here a putative anchoring factor (unknown) binds to the Xist A-repeat (Rep-A) and interacts with the polymerized SAF-A/hnRNPU scaffold. The factor functions additively with CIZ1 which is present and bound to Xist RNA in all cell types. The overall effect is an increased strength of Xist RNA anchoring relative to (*a*), where only CIZ1 is present.

maps showing sites of Xist RNA enrichment across the entire length of the chromosome. In the study by Engreitz *et al.* [27] an inducible promoter was used to express Xist RNA in ES cells. At early time points, 1–3 h, Xist RNA accumulation was found to occur preferentially at a small number of hotspots. Further analysis demonstrated that the preferred sites correspond to chromosome regions that have high contact frequencies with the Xist transcription site, as determined by Hi-C mapping [56]. Subsequently Xist RNA was observed to spread to other sites with relatively high contact frequencies, eventually accumulating widely over the entire chromosome. In light of these findings, it is interesting to consider that relative enrichment of Xist RNA over gene-rich regions could be related to chromosome compartments where gene-rich domains (A-type compartments) interact together more than with gene-poor domains (B-type compartments), and vice versa [56].

The study by Engreitz *et al.* [27] further noted a relatively slow accumulation of Xist RNA over regions containing genes that are strongly expressed in ES cells and, moreover, that the eventual spread of Xist RNA to these regions is dependent on gene silencing (using a mutant Xist RNA lacking the critical A-repeat silencing element). A relationship between the rate of accumulation of Xist RNA over regions dependent on gene activity was also found by Simon *et al.* [28] using CHART-seq, with the observed rate of Xist RNA accumulation being slower in differentiating ES cells that are establishing X inactivation than in MEF cells in which Xist RNA was transiently depleted by treatment with locked nucleic acid (LNA) antisense probes. One suggestion for this observation is that the relatively compacted Xi chromatin defines a smaller volume for the spread of newly transcribed Xist RNA.

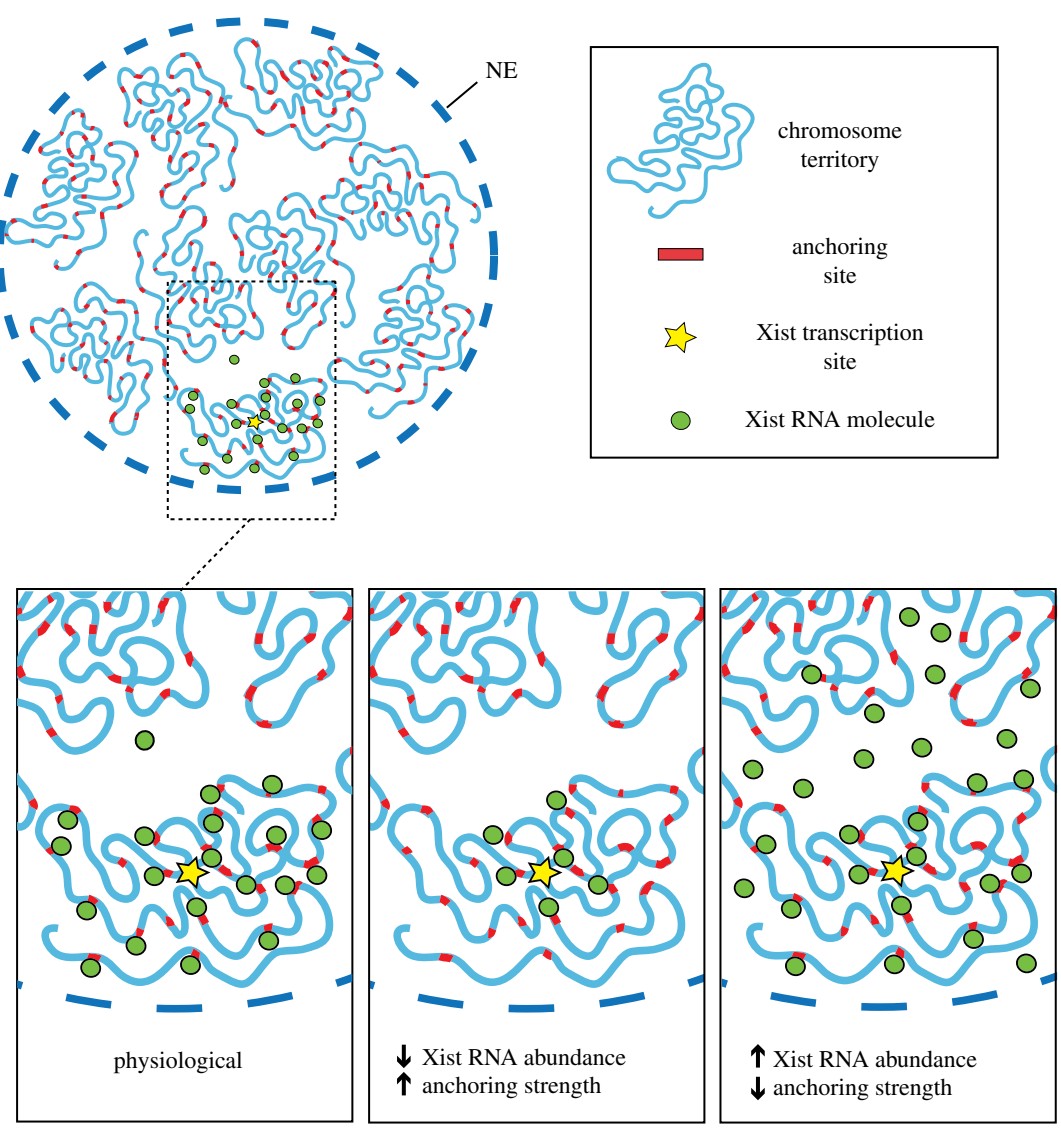

**Figure 3.** A simplified model for localized accumulation of Xist RNA. Depiction of an interphase nucleus (NE = nuclear envelope), with individual chromosomes indicated with a fixed number of putative anchoring sites (top). In the expanded view (bottom left panel) the Xi chromosome is shown with the majority of Xist RNA molecules anchored at perichromatin sites (left panel). The model proposes that Xist RNA accumulation is centred on the site of synthesis with the range being determined by the abundance (product of synthesis and turnover rates) and the strength of anchoring. The number of putative anchoring sites is considered as constant. Decreasing abundance or increasing anchoring strength are predicted to reduce the local accumulation range from the site of synthesis (bottom centre panel). Conversely, increasing abundance or reducing anchoring strength are predicted to result in expanded Xist RNA accumulation beyond the Xi territory (bottom right panel).

Taken together the RAP-seq and CHART-seq studies indicate that chromosome topology/dynamics and the progression of Xist-mediated silencing are important in determining the location of Xist RNA-anchoring sites. Whether or not this is sufficient to explain the limited capacity of a sister chromatid for anchoring Xist RNA remains to be determined.

## 5. Modelling localized accumulation of Xist RNA

With our increased knowledge of the factors required for localized accumulation of Xist RNA and the consequence of their depletion we can begin to formulate testable models. In the simplest case we can conjecture that Xist RNA molecules diffuse away from the site of transcription, with their spread being held in check through anchoring interactions with proteins of the nuclear matrix. In this model the key parameters that will define the range or volume over which Xist RNA

accumulates are the dynamics of the nuclear matrix interaction ($K_{on}/K_{off}$ and dwell time), the concentration of Xist RNA (the product of the rate of synthesis and degradation) and the abundance of Xist RNA-anchoring sites (figure 3). This model provides a basis to explain how during sex chromosome evolution progressive enlargement of the domain of gene silencing that accompanied the stepwise attrition of the sex-determining Y chromosome [57] may have occurred through increasing the range over which Xist RNA accumulates in relation to its site of synthesis.

Rigorous testing of the aforementioned model will require accurate determination of the key parameters. Counting single Xist RNA molecules in super-resolution microscopy experiments leads to estimates of around 100–200 molecules/cell, approximating to one Xist molecule/Mb of silenced chromatin [19,20]. Estimates of Xist RNA turnover (half-life), based on decay following transcriptional inhibition, are in the range of 2–6 h [58,59]. However, a study that pioneered live cell imaging of Xist RNA [55] suggested that Xist turnover is

royalsocietypublishing.org/journal/rsob Open Biol. 9: 190213

accelerated considerably by ongoing transcription, with reported half-life values, based on fluorescence recovery after photobleaching (FRAP), being around 3–4 h in the presence of the transcriptional inhibitor actinomycin B and around 30 min in its absence. Further studies are required both to confirm the aforementioned values and to establish additional parameters, notably the kinetics of Xist RNA anchoring.

While meaningful validation of the above model requires further analysis, there is some supporting evidence worthy of discussion. Thus, the model predicts that weakening of the anchoring interaction will result in dispersal of Xist RNA molecules (figure 3), as is observed in SAF-A/hnRNPU and CIZ1 loss of function experiments. It should be noted that the fact that Xist molecules disperse only within the nucleoplasm and are never seen to reach the cytoplasm suggests that Xist RNA, despite being a spliced and polyadenylated RNAPII transcript, has evolved to evade mRNA export pathways. Indeed Xist transcripts show very similar behaviour when transcribed from intronless transgenes [60]. The molecular basis for this atypical property of Xist RNA is presently unknown, although there is evidence indicating that human XIST RNA is associated with relatively low levels of the mRNA export factor TAP/Nxf1 [61]. Further investigation of how Xist RNA evades nuclear export will be an interesting avenue for future studies.

The above model predicts that the range of Xist RNA spreading away from the site of transcription should scale with Xist RNA abundance and stability (figure 3), and here there is also some supporting evidence. One case comes from the study of balanced X: autosome translocations in which autosomal regions that are *in cis* with an Xist-bearing X chromosome segment exhibit Xist-mediated silencing that is often attenuated with distance from the translocation breakpoint [62]. One such translocation, characterized in the laboratory mouse, is T(X;4)37H (T37H), in which the translocation product bearing the Xist gene is extremely large, representing most of the X chromosome and the relatively large autosome, chromosome 4. Classical genetic studies demonstrated attenuated spread of X inactivation occurring from the breakpoint into chromosome 4 [63]. Interestingly, molecular analysis of Xist RNA distribution on T37H chromosomes at metaphase also demonstrated attenuated spread [64]; importantly, because the experiments were conducted at the onset of X inactivation in differentiating XX ES cells, the observation indicates limited spread of Xist RNA rather than spread followed by retreat. Thus, in this example the increased size of the Xist-expressing chromosome, and potentially the number of Xist RNA-anchoring sites relative to Xist RNA molecules, could explain the limited range of spread along the translocated chromosome 4 sequences.

An additional example supporting a relationship between range and RNA concentration has come in a recent study analysing the imprinted lncRNAs Kcnq1ot1 and Airn [65], which, as discussed above, are thought to function similarly to Xist RNA. Specifically it was reported that in mouse trophoblast stem cells both Kcnq1ot1 and Airn recruit Polycomb repressive complexes to modify underlying chromatin over large (10–20 Mb) domains; moreover, like Xist RNA, this pathway depends on the RBP hnRNPK [65,66]. Interestingly, estimation of the abundance of Kcnq1ot1 and Airn RNAs ranged from 10 to 20 molecules/cell, around 10-fold lower than estimates of Xist RNA abundance [65]. Given that Airn and Kcnq1ot1 RNAs influence chromatin modification states within contiguous domains around 1/10th of the size of the X chromosome,

these examples serve to further exemplify the apparent relationship between the range of action of *cis*-acting lncRNAs and their abundance/stability.

It should be noted that the aforementioned model makes a number of untested assumptions. One notable example is that Xist RNA molecules translocate away from the site of transcription by passive diffusion, being retarded by anchoring to the nuclear matrix. In one scenario individual Xist RNA molecules could undergo several cycles of anchor and release. Alternatively, each molecule may undergo only a single anchoring interaction with a duration that is of a similar order to the lifespan of individual Xist RNA molecules (figure 4a). However, Xist RNPs are large, estimated to be around 40–50 $nm^3$, while the interchromatin spaces in the Xi territory through which Xist RNA must diffuse appear to be relatively narrow [20]. Thus, it is plausible that Xist RNA molecules require facilitated diffusion or active transport in order to translocate within this spatially restricted environment. An extreme possibility is that Xist RNA molecules translocate from one site to another through a series of jumps mediated by dynamic chromatin/chromosome contacts, analogous to the transfer of a Velcro ball when two Velcro bats are brought together (illustrated in figure 4b). This model integrates the concept that preferred anchoring sites correlate with chromosomal regions that have relatively high contact frequencies with the site of Xist RNA synthesis.

## 6. Future perspectives

Recent progress has allowed us to begin to formulate ideas for how Xist RNA accumulates locally over the chromosome from which it is transcribed. There are however several significant gaps in our understanding; moreover, there is a need to develop better tools and approaches to test emergent ideas. Here I highlight key issues for future work in this field.

SAF-A/hnRNPU and CIZ1 are clearly not the only factors involved in Xist RNA anchoring, and similarly the Xist E-repeat, to which CIZ1 binds, is not the only important sequence element. Indeed, an early study that analysed the effect of multiple deletions in Xist sequence concluded that there is no single element in Xist RNA required for localized accumulation, but rather multiple elements that function redundantly [54]. The work on CIZ1, and other studies, highlights that localized accumulation mechanisms vary in different cell types and developmental stages [44,51]. Defining the missing factors and Xist sequence elements is therefore an important priority. There are some useful clues: in relation to key Xist RNA elements, the C-repeat in mouse Xist RNA has been suggested to contribute to localized accumulation through experiments in which antisense LNA/peptide nucleic acid oligonucleotides were shown to expel locally accumulated Xist RNA in MEF cell lines [67,68]. Additionally, the Xist A-repeat was shown to be required for localized accumulation of a truncated Xist transgene which spans the first 4 kb of the Xist sequence [54]. More recently deletion of the Xist B repeat, through which Polycomb repressors are recruited via hnRNPK binding [66,69], was found to disrupt localized accumulation in MEFs [70]. Finally, a recent report has documented defects in localized accumulation of Xist RNA in ES cells in which the Xist A-repeat is deleted [39], consistent with the aforementioned finding that the A-repeat is required for

**8**

royalsocietypublishing.org/journal/rsob    *Open Biol.* **9**: 190213

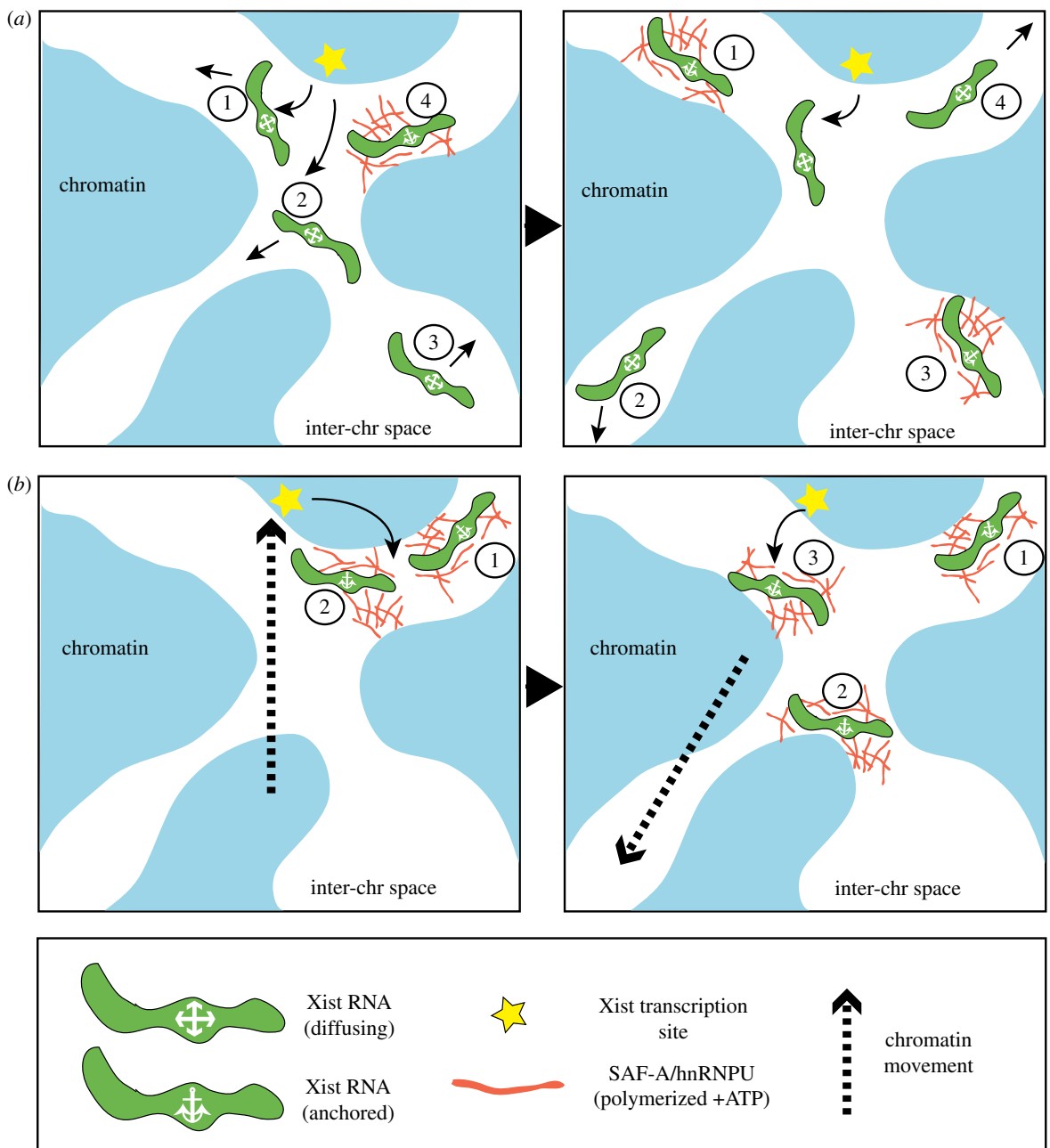

**Figure 4.** Models for the translocation of Xist RNA molecules. (*a*) Individual Xist RNA molecules diffuse away from the transcription site, moving through inter-chromatin (inter-chr) spaces in directions indicated with arrows (left panel, molecules labelled 1–3). Individual molecules are restrained at defined sites through interactions between polymerized SAF-A/hnRNPU and Xist-anchoring factors as proposed in figure 2 (left panel, molecule labelled 4). Non-anchored molecules diffuse further and then either anchor at distant sites (right panel, molecules 1 and 3) or continue to diffuse (right panel, molecule 2). Depending on anchoring dynamics (dwell time) and RNA turnover, Xist RNA molecules may undergo cycles of anchor release (right panel, molecule 4). (*b*) An alternative model for the translocation of Xist RNA molecules. Xist RNA is anchored in close proximity to the transcription site (left panel, molecules 1 and 2) and dynamic movements of the chromatin (dashed arrow) transfer anchored molecules from one site to another (right panel, molecule 2). Other newly transcribed Xist molecules then anchor (right panel, molecule 3) and can undergo translocation to alternative sites as a result of different dynamic chromatin movements.

localized accumulation in the context of a truncated Xist transgene [54]. The former study noted aberrant localized accumulation in cells lacking the key silencing factor SPEN, which binds to the Xist A-repeat [39].

An important caveat in considering the role of the A- and B-repeats and associated factors in localized accumulation of Xist RNA is that the mutations described also disrupt gene silencing and Xi chromatin modification. Thus, it is plausible that aberrant localized accumulation deficiencies are a consequence of deficiencies in silencing/chromatin modification, for example reflecting a relatively decompacted chromosome structure. In future studies it will be important to test this by

attempting to define separation of function mutations in the key A- and B-repeat-associated factors.

Identifying the factors and elements with a role in localized accumulation of Xist RNA will not in itself explain the underlying mechanisms. For this it will be necessary to accurately measure key parameters: localization, anchoring/diffusion dynamics and turnover of Xist RNA. Methods such as RAP-seq and CHART-seq allow high-resolution mapping of Xist localization and can also provide limited temporal information over long time courses, several hours to days. However, the data are an aggregate from many millions of cells and it is difficult to discern from them

events at the single-cell level. It may be possible in the future to develop single-cell RAP-seq/CHART-seq, although this will require highly efficient capture of Xist–chromatin complexes. Also, a drawback with these methods is that they rely on efficient cross-linking of Xist RNA with neighbouring chromatin, which could introduce bias towards Xist RNA sites that are in closer proximity to chromatin. A complementary strategy to advance our understanding of local accumulation of Xist RNA is to further develop the application of super-resolution microscopy to define the organization of Xist RNA particles relative to other nuclear landmarks and structures, in either fixed or cryopreserved material. One option would be to apply correlative light and electron microscopy, which should provide details on the ultrastructural organization of Xist particles relative to chromatin and other nuclear landmarks.

While methods based on analysing fixed/cryopreserved cells should be highly informative, analysis of Xist RNA dynamics ultimately requires the development of live-cell imaging approaches. In experiments to date, this has been achieved by labelling Xist RNA with green fluorescent protein using the MS2 aptamer system [55]. In future it will be important to develop approaches that allow the movements of individual Xist RNA molecules to be followed in real time. Achieving sufficiently fast imaging at high resolution while minimizing phototoxicity will prove challenging, but has the potential to answer the key outstanding questions regarding how Xist RNA particles are transferred along the length of an entire chromosome.

The aforementioned approaches, using either fixed cells or live-cell imaging, will be more informative when coupled to perturbation experiments in which key factors/sequence elements are deleted/ablated. Here again there may be significant challenges, for example SAF-A/hnRNPU is required for viability in ES cells. Accordingly it will be important to make full use of systems for conditional knockout, using either inducible gene deletion or protein degradation-based strategies.

A final and complementary approach to unravelling the mechanism for localized accumulation of Xist RNA will be to develop synthetic RNAs that can bind key factors, for example the anchoring protein CIZ1, using established RNA aptamer systems, such as MS2 [55,71], BglG [72] or λBoxB [73]. Modulation of the number of binding sites and/or the level and stability of the RNA could then be used to test emergent models and define the minimum requirements for localized accumulation to occur. Additionally, developing approaches to regulate the level or stability of Xist RNA will be useful for probing the relationship between Xist RNA concentration and its range of action.

# 7. Concluding comments

The unique and unusual property of Xist RNA to accumulate locally over the chromosome from which it is transcribed provides us with an intriguing and challenging mystery. From a position of near-complete ignorance we can now begin to identify the underlying mechanisms. The coming years are set to be exciting as we move towards completing the task of defining the key factors and of testing our developing models.

Data accessibility. This article has no additional data.

Competing interests. I declare I have no competing interests.

Funding. N.B. is funded by Wellcome (grant no. 215513/Z/19/Z).

Acknowledgements. I would like to thank colleagues in the Brockdorff laboratory for stimulating discussions and Heather Coker and Tatyana Nesterova for providing microscopy images. N.B. is a Wellcome Trust Principal Research Fellow (grant no. 103768).

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
