## [Reviewer comments · Open Biology]

Review History

RSOB-19-0213.R0 (Original submission)

Review form: Reviewer 1 (Jeanne Lawrence)

Recommendation

Accept with minor revision (please list in comments)

Do you have any ethical concerns with this paper?

No

Comments to the Author

This is a focused review on a topic of significant interest, regarding how Xist RNA localizes to the inactive X chromosome in cis. While there have been many reviews of X-chromosome inactivation, they mostly focus on other aspects of the X-chromosome inactivation process, such as Xist transcriptional regulation and/or the various histone/chromatin modifications that XIST RNA triggers to enact silencing. Hence, the distinctive focus of this review makes it a stronger contribution and avoids being “just another” review of XCI, and . This review from a leader in the field works to connect several different observations to support a model for how Xist RNA comes to localize to the chromosome from which it is transcribed, and discusses other similar

lncRNAs which “localize” to nuclear chromosomes in cis. The model proposes that Xist RNA diffuses from the transcription site through interchromatin channels but becomes anchored by triggering ATP-dependent polymerization of hnRNP-U/SAF-A, which in turn associates with CIZ1. It is suggested that Xist transcripts may go through cycles of anchor and release, or anchor once but translocate to other sites via chromatin movement. A major point of the model is that Xist preferentially binds gene-rich chromatin, excluded from gene-poor.

With some exceptions this review does a reasonable job of summarizing the literature and provide thought-provoking suggestions into the fascinating question of how Xist transcripts spread across a chromosome. I appreciate the effort to envision how “matrix” proteins can contribute to this, although I am not necessarily persuaded by some main aspects of the model, it was intended to suggest future avenues of experimentation, which it does.

Several points would benefit from revision and are intended to be constructive, as listed below.

1) A fundamental premise of the review is that Xist RNA localizes in cis like a “cloud” over the chromosome territory, and nuclear accumulations of several other lncRNAs are mentioned as indicative of function. However, nuclear accumulations are seen for most expressed RNAs, and can be rather large for a variety of reasons, that are not indicative of function on chromatin. The evidence that XIST RNA had a unique relationship coating the interphase nuclear chromosome structure (and detaches at mitosis) came from Clemson et al (J Cell Bio, 1996) (not from the papers cited). Clemson is cited for a later point, but it is recommended to relook at this study as it demonstrates several points relevant to the review that the author may not realize. It also illustrates the types of analyses required to discriminate unusual chromatin-association from the normal accumulation of an RNA at the site of synthesis/processing, and more awareness of this in the field would be useful. In referring to other localized nuclear RNAs presumed to have similarity to Xist RNA, it would be helpful to state whether they had clearly been demonstrated to morphologically associate with chromatin beyond being transcribed and spliced there.

2) The one photograph shown in Fig 1 doesn't in my view provide a useful or accurate depiction of what Xist RNA FISH looks like with conventional microscopy, nor is it clear that the “super-resolution” photomicrograph is a robust detection of Xist RNA. The standard fluorescence image appears poorly focused, and could have been optically sectioned and deconvolved to provide much more precise detail and punctate distribution. It also appears to represent less RNA signal as well as less resolution than a good hybridization and image. The 3D-Sim image also looks questionable because it may have started with a weak or incomplete signal, and one wonders whether sensitivity of detection (fading) is sacrificed for resolution (as is a common issue with super-resolution approaches). But for any visualization of RNA in cells, the most important things are the preservation of the RNA, the accessibility for hybridization, the hybridization efficiency and the detection of hybridization. If those steps are sub-par, one won't be able to make a silk purse..... . This may not seem like a big point, but readers may take the message that one must use super-resolution procedures to have a strong visualization of Xist RNA FISH, or that any visualization by a high-powered microscope is the “real” result, whereas it could be quite limited by prior steps (or rapid fading). The Figure should include a good, strong high-resolution standard images of Xist RNA in cells.

3) Central to the authors model is that Xist transcripts are in channels within the X-chromosome territory, relying on limited reports of this very detailed molecular-level relationship of Xist molecules to DNA and certain histone modifications and “channels”. This comes across as a definitive finding, but the authors might consider that this level of detail is very difficult to preserve, detect, and visualize fully, even by higher-resolution fluorescence microscopy.

4) The author states that Xist RNA is “excluded” from gene-poor DNA and cites Duthie, '99 (for FISH), and Marks '09 and Pinter, '12 (for molecular), but other evidence indicates that Xist does coat gene-poor regions. KP Smith et al. (2004) found that Xist RNA localized more homogeneously across the chromosome (except the centromere) earlier in mitosis, and then

released from gene-rich bands last, indicating this was a timing difference, not a lack of binding in gene-poor regions. Engreitz et al (2013) and Simon (2013), looking at the molecular level also found that Xist RNA is on gene-poor DNA (already silent, repeat-rich DNA and G-bands). Also, recent reports by Zyllicz et al '18 further suggest gene-poor chromatin is the first to acquire repressive PcG marks (or loss of active marks) during XCI.

5) The legend or point of Figure 2 A versus B wasn't very clear.....what is the distinction here, whether the model focuses on RepA or not? The Figure 3 concept was clear enough after rechecking the text description (cycles of anchoring versus single anchoring and then chromatin movement) but the figure itself was not easily digested. A model that showed an alternative to "channels", rather than so many modifications of the channel concept, might have been a more helpful option.

6) The model also doesn't show the chromosome territory, and seems to avoid the question of how Xist is limited to that chromosome? Am I correct that the implication is that the spread would be limited by the amount of Xist?

7) If memory serves correctly, I believe the Gilbert paper concluded that the ATP dependent oligomerization of SAF-A was linked to active transcription and open euchromatin, and I believe may have indicated that SAF-A was not on heterochromatin. In any case, is this the opposite of what is proposed here, where polymerization closes rather than opens chromatin?

Review form: Reviewer 2

Recommendation

Accept with minor revision (please list in comments)

Do you have any ethical concerns with this paper?

No

Comments to the Author

The review "Localised accumulation of Xist RNA in X chromosome inactivation" describes recent findings in X chromosome inactivation. In particular, the following important question is considered in details: how Xist RNA accumulates locally on the chromosome from which it is transcribed. The author suggest several models of localised accumulation of Xist RNA in cis accompanied by schemes. The manuscript provides an important contribution to the field, is well-written and comprehensive.

However some questions about the described mechanisms of Xist RNA retention in the chromatin of inactive X chromosome arise:

1) It is stated that "Xist RNA is anchored through interactions with proteins of the nuclear matrix". However, interaction of Xist RNA with SAF-A/hnRNPU can be also explained by co-transcriptional binding of this protein to nascent transcript and formation of a stable RNP particle. It is also known that SAF-A/hnRNPU could have a prion-like activity and thus form a biomolecular condensate leading to formation of dynamic membraneless compartments. Thus it is better to use this modern terminology instead of rigid "nuclear matrix".

2) Chromosome conformation capture approaches revealed global loss of local structure on the inactive X chromosome and formation of large mega-domains. I suggest to discuss the possibility

of preferential spreading of Xist RNA along inactive X chromosome due to the absence of such local structure including active/inactive compartments and topologically associating domains with their insulating properties.

Decision letter (RSOB-19-0213.R0)

07-Oct-2019

Dear Dr Brockdorff

We are pleased to inform you that your manuscript RSOB-19-0213 entitled "Localised accumulation of Xist RNA in X chromosome inactivation" has been accepted by the Editor for publication in Open Biology. The reviewer(s) have recommended publication, but also suggest some minor revisions to your manuscript. Therefore, we invite you to respond to the reviewer(s)' comments and revise your manuscript.

Please submit the revised version of your manuscript within 14 days. If you do not think you will be able to meet this date please let us know immediately and we can extend this deadline for you.

- 1) A text file of the manuscript (doc, txt, rtf or tex), including the references, tables (including captions) and figure captions. Please remove any tracked changes from the text before submission. PDF files are not an accepted format for the "Main Document".
- 2) A separate electronic file of each figure (tiff, EPS or print-quality PDF preferred). The format should be produced directly from original creation package, or original software format. Please note that PowerPoint files are not accepted.
- 3) Electronic supplementary material: this should be contained in a separate file from the main text and meet our ESM criteria (see <http://royalsocietypublishing.org/instructions-authors#question5>). All supplementary materials accompanying an accepted article will be treated as in their final form. They will be published alongside the paper on the journal website and posted on the online figshare repository. Files on figshare will be made available

approximately one week before the accompanying article so that the supplementary material can be attributed a unique DOI.

Online supplementary material will also carry the title and description provided during submission, so please ensure these are accurate and informative. Note that the Royal Society will not edit or typeset supplementary material and it will be hosted as provided. Please ensure that the supplementary material includes the paper details (authors, title, journal name, article DOI). Your article DOI will be 10.1098/rsob.2016[last 4 digits of e.g. 10.1098/rsob.20160049].

4) A media summary: a short non-technical summary (up to 100 words) of the key findings/importance of your manuscript. Please try to write in simple English, avoid jargon, explain the importance of the topic, outline the main implications and describe why this topic is newsworthy.

Data-Sharing

It is a condition of publication that data supporting your paper are made available. Data should be made available either in the electronic supplementary material or through an appropriate repository. Details of how to access data should be included in your paper. Please see <http://royalsocietypublishing.org/site/authors/policy.xhtml#question6> for more details.

Data accessibility section

Sincerely,
The Open Biology Team
<mailto:openbiology@royalsociety.org>

Reviewer(s)' Comments to Author:

Referee: 1

Comments to the Author(s)

This is a focused review on a topic of significant interest, regarding how Xist RNA localizes to the inactive X chromosome in cis. While there have been many reviews of X-chromosome inactivation, they mostly focus on other aspects of the X-chromosome inactivation process, such as Xist transcriptional regulation and/or the various histone/chromatin modifications that XIST RNA triggers to enact silencing. Hence, the distinctive focus of this review makes it a stronger contribution and avoids being "just another" review of XCI, and . This review from a leader in the field works to connect several different observations to support a model for how Xist RNA comes to localize to the chromosome from which it is transcribed, and discusses other similar lncRNAs which "localize" to nuclear chromosomes in cis. The model proposes that Xist RNA diffuses from the transcription site through interchromatin channels but becomes anchored by triggering ATP-dependent polymerization of hnRNP-U/SAF-A, which in turn associates with

CIZ1. It is suggested that Xist transcripts may go through cycles of anchor and release, or anchor once but translocate to other sites via chromatin movement. A major point of the model is that Xist preferentially binds gene-rich chromatin, excluded from gene-poor.

With some exceptions this review does a reasonable job of summarizing the literature and provide thought-provoking suggestions into the fascinating question of how Xist transcripts spread across a chromosome. I appreciate the effort to envision how “matrix” proteins can contribute to this, although I am not necessarily persuaded by some main aspects of the model, it was intended to suggest future avenues of experimentation, which it does.

Several points would benefit from revision and are intended to be constructive, as listed below.

1) A fundamental premise of the review is that Xist RNA localizes in cis like a “cloud” over the chromosome territory, and nuclear accumulations of several other lncRNAs are mentioned as indicative of function. However, nuclear accumulations are seen for most expressed RNAs, and can be rather large for a variety of reasons, that are not indicative of function on chromatin. The evidence that XIST RNA had a unique relationship coating the interphase nuclear chromosome structure (and detaches at mitosis) came from Clemson et al (J Cell Bio, 1996) (not from the papers cited). Clemson is cited for a later point, but it is recommended to relook at this study as it demonstrates several points relevant to the review that the author may not realize. It also illustrates the types of analyses required to discriminate unusual chromatin-association from the normal accumulation of an RNA at the site of synthesis/processing, and more awareness of this in the field would be useful. In referring to other localized nuclear RNAs presumed to have similarity to Xist RNA, it would be helpful to state whether they had clearly been demonstrated to morphologically associate with chromatin beyond being transcribed and spliced there.

2) The one photograph shown in Fig 1 doesn't in my view provide a useful or accurate depiction of what Xist RNA FISH looks like with conventional microscopy, nor is it clear that the “super-resolution” photomicrograph is a robust detection of Xist RNA. The standard fluorescence image appears poorly focused, and could have been optically sectioned and deconvolved to provide much more precise detail and punctate distribution. It also appears to represent less RNA signal as well as less resolution than a good hybridization and image. The 3D-Sim image also looks questionable because it may have started with a weak or incomplete signal, and one wonders whether sensitivity of detection (fading) is sacrificed for resolution (as is a common issue with super-resolution approaches). But for any visualization of RNA in cells, the most important things are the preservation of the RNA, the accessibility for hybridization, the hybridization efficiency and the detection of hybridization. If those steps are sub-par, one won't be able to make a silk purse..... . This may not seem like a big point, but readers may take the message that one must use super-resolution procedures to have a strong visualization of Xist RNA FISH, or that any visualization by a high-powered microscope is the “real” result, whereas it could be quite limited by prior steps (or rapid fading). The Figure should include a good, strong high-resolution standard images of Xist RNA in cells.

3) Central to the authors model is that Xist transcripts are in channels within the X-chromosome territory, relying on limited reports of this very detailed molecular-level relationship of Xist molecules to DNA and certain histone modifications and “channels”. This comes across as a definitive finding, but the authors might consider that this level of detail is very difficult to preserve, detect, and visualize fully, even by higher-resolution fluorescence microscopy.

4) The author states that Xist RNA is “excluded” from gene-poor DNA and cites Duthie, '99 (for FISH), and Marks '09 and Pinter, '12 (for molecular), but other evidence indicates that Xist does coat gene-poor regions. KP Smith et al. (2004) found that Xist RNA localized more homogeneously across the chromosome (except the centromere) earlier in mitosis, and then released from gene-rich bands last, indicating this was a timing difference, not a lack of binding in gene-poor regions. Engreitz et al (2013) and Simon (2013), looking at the molecular level also found that Xist RNA is on gene-poor DNA (already silent, repeat-rich DNA and G-bands). Also,

recent reports by Zyllicz et al '18 further suggest gene-poor chromatin is the first to acquire repressive PcG marks (or loss of active marks) during XCI.

5) The legend or point of Figure 2 A versus B wasn't very clear.....what is the distinction here, whether the model focuses on RepA or not? The Figure 3 concept was clear enough after rechecking the text description (cycles of anchoring versus single anchoring and then chromatin movement) but the figure itself was not easily digested. A model that showed an alternative to "channels", rather than so many modifications of the channel concept, might have been a more helpful option.

6) The model also doesn't show the chromosome territory, and seems to avoid the question of how Xist is limited to that chromosome? Am I correct that the implication is that the spread would be limited by the amount of Xist?

7) If memory serves correctly, I believe the Gilbert paper concluded that the ATP dependent oligomerization of SAF-A was linked to active transcription and open euchromatin, and I believe may have indicated that SAF-A was not on heterochromatin. In any case, is this the opposite of what is proposed here, where polymerization closes rather than opens chromatin?

Referee: 2

Comments to the Author(s)

The review "Localised accumulation of Xist RNA in X chromosome inactivation" describes recent findings in X chromosome inactivation. In particular, the following important question is considered in details: how Xist RNA accumulates locally on the chromosome from which it is transcribed. The author suggest several models of localised accumulation of Xist RNA in cis accompanied by schemes. The manuscript provides an important contribution to the field, is well-written and comprehensive.

However some questions about the described mechanisms of Xist RNA retention in the chromatin of inactive X chromosome arise:

1) It is stated that "Xist RNA is anchored through interactions with proteins of the nuclear matrix". However, interaction of Xist RNA with SAF-A/hnRNPU can be also explained by co-transcriptional binding of this protein to nascent transcript and formation of a stable RNP particle. It is also known that SAF-A/hnRNPU could have a prion-like activity and thus form a biomolecular condensate leading to formation of dynamic membraneless compartments. Thus it is better to use this modern terminology instead of rigid "nuclear matrix".

2) Chromosome conformation capture approaches revealed global loss of local structure on the inactive X chromosome and formation of large mega-domains. I suggest to discuss the possibility of preferential spreading of Xist RNA along inactive X chromosome due to the absence of such local structure including active/inactive compartments and topologically associating domains with their insulating properties.

Author's Response to Decision Letter for (RSOB-19-0213.R0)

See Appendix A.

Decision letter (RSOB-19-0213.R1)

04-Nov-2019

Dear Dr Brockdorff

We are pleased to inform you that your manuscript entitled "Localised accumulation of Xist RNA in X chromosome inactivation" has been accepted by the Editor for publication in Open Biology.

Sincerely,

The Open Biology Team
mailto: openbiology@royalsociety.org

Appendix A

I would like to thank the reviewers for their constructive comments and suggestions. I have modified and improved the manuscript taking these into account as detailed below (my comments in italics).

Reviewer(s)' Comments to Author:

Referee: 1

Comments to the Author(s)

This is a focused review on a topic of significant interest, regarding how Xist RNA localizes to the inactive X chromosome in cis. While there have been many reviews of X-chromosome inactivation, they mostly focus on other aspects of the X-chromosome inactivation process, such as Xist transcriptional regulation and/or the various histone/chromatin modifications that XIST RNA triggers to enact silencing. Hence, the distinctive focus of this review makes it a stronger contribution and avoids being “just another” review of XCI, and . This review from a leader in the field works to connect several different observations to support a model for how Xist RNA comes to localize to the chromosome from which it is transcribed, and discusses other similar lncRNAs which “localize” to nuclear chromosomes in cis. The model proposes that Xist RNA diffuses from the transcription site through interchromatin channels but becomes anchored by triggering ATP-dependent polymerization of hnRNP-U/SAF-A, which in turn associates with CIZ1. It is suggested that Xist transcripts may go through cycles of anchor and release, or anchor once but translocate to other sites via chromatin movement. A major point of the model is that Xist preferentially binds gene-rich chromatin, excluded from gene-poor.

With some exceptions this review does a reasonable job of summarizing the literature and provide thought-provoking suggestions into the fascinating question of how Xist transcripts spread across a chromosome. I appreciate the effort to envision how “matrix” proteins can contribute to this, although I am not necessarily persuaded by some main aspects of the model, it was intended to suggest future avenues of experimentation, which it does.

Several points would benefit from revision and are intended to be constructive, as listed below.

1) A fundamental premise of the review is that Xist RNA localizes in cis like a “cloud” over the chromosome territory, and nuclear accumulations of several other lncRNAs are mentioned as indicative of function. However, nuclear accumulations are seen for most expressed RNAs, and can be rather large for a variety of reasons, that are not indicative of function on chromatin. The evidence that XIST RNA had a unique relationship coating the interphase nuclear chromosome structure (and detaches at mitosis) came from Clemson et al (J Cell Bio, 1996) (not from the papers cited). Clemson is cited for a later point, but it is recommended to relook at this study as it demonstrates several points relevant to the review that the author may not realize. It also illustrates the types of analyses required to discriminate unusual chromatin-association from the normal accumulation of an RNA at the site of synthesis/processing, and more awareness of this in the field would be useful. In referring to other localized nuclear RNAs presumed to have similarity to Xist RNA, it would be helpful to state whether they had clearly been demonstrated to morphologically associate with chromatin beyond being transcribed and spliced there.

The citation has been corrected as requested (ref 8, p3 and elsewhere). Additional comments have been added to highlight the basis for discriminating Xist RNA from other RNAs (p3). Discussion of other potentially related RNAs has been modified to make clear that further analysis is required to determine exactly which properties, for example nuclear matrix association, are shared.

2) The one photograph shown in Fig 1 doesn't in my view provide a useful or accurate depiction of what Xist RNA FISH looks like with conventional microscopy, nor is it clear that the “super-resolution” photomicrograph is a robust detection of Xist RNA. The standard fluorescence

image appears poorly focused, and could have been optically sectioned and deconvolved to provide much more precise detail and punctate distribution. It also appears to represent less RNA signal as well as less resolution than a good hybridization and image. The 3D-Sim image also looks questionable because it may have started with a weak or incomplete signal, and one wonders whether sensitivity of detection (fading) is sacrificed for resolution (as is a common issue with super-resolution approaches). But for any visualization of RNA in cells, the most important things are the preservation of the RNA, the accessibility for hybridization, the hybridization efficiency and the detection of hybridization. If those steps are sub-par, one won't be able to make a silk purse..... This may not seem like a big point, but readers may take the message that one must use super-resolution procedures to have a strong visualization of Xist RNA FISH, or that any visualization by a high-powered microscope is the "real" result, whereas it could be quite limited by prior steps (or rapid fading). The Figure should include a good, strong high-resolution standard images of Xist RNA in cells.

As suggested the image in Figure 1 has been replaced with better representative examples of Xist RNA FISH using either conventional widefield or super-resolution 3D-SIM. The purpose of showing both is to highlight that super-resolution imaging can provide additional insights into Xist behaviour. In fact the sample preparation used for 3D-SIM is very similar and if anything more gentle than that used in conventional RNA FISH experiments. This contrasts with PALM/STORM imaging experiments published by others where disruption is more significant.

3) Central to the authors model is that Xist transcripts are in channels within the X-chromosome territory, relying on limited reports of this very detailed molecular-level relationship of Xist molecules to DNA and certain histone modifications and "channels". This comes across as a definitive finding, but the authors might consider that this level of detail is very difficult to preserve, detect, and visualize fully, even by higher-resolution fluorescence microscopy.

There is no question that chromatin at interphase is surrounded by inter-chromatin spaces and the data defining the relationship of Xist, nuclear matrix proteins and chromatin is in my view accurate and robust. As above, the conditions used to fix cells in these experiments were if anything more gentle than conventional methods for fluorescence microscopy. I appreciate that the term 'channel' could be viewed as leading towards a particular interpretation and have therefore revised the text and relevant Figures using 'interchromatin space' instead of channel.

4) The author states that Xist RNA is "excluded" from gene-poor DNA and cites Duthie, '99 (for FISH), and Marks '09 and Pinter, '12 (for molecular), but other evidence indicates that Xist does coat gene-poor regions. KP Smith et al. (2004) found that Xist RNA localized more homogeneously across the chromosome (except the centromere) earlier in mitosis, and then released from gene-rich bands last, indicating this was a timing difference, not a lack of binding in gene-poor regions. Engreitz et al (2013) and Simon (2013), looking at the molecular level also found that Xist RNA is on gene-poor DNA (already silent, repeat-rich DNA and G-bands). Also, recent reports by Zyllicz et al '18 further suggest gene-poor chromatin is the first to acquire repressive PcG marks (or loss of active marks) during XCI.

I have modified the text (p6) to give a more balanced view of the question of whether Xist preferentially associates with gene-rich domains throughout the cell cycle. I did not include reference to Zyllicz et al 2018 as this study does not directly measure Xist localisation on the X chromosome. One argument that I didn't go into is that large constitutive heterochromatin blocks on the X chromosome in vole were shown to exclude Xist RNA throughout the cell cycle (Duthie et al study), and based on this and Xist RNA exclusion from centromeres, I believe that it is likely generalizable that gene-poor regions are depleted for Xist RNA. Apparently conflicting findings based on HTS

methodologies must be caveated by the fact that large repeat sequence blocks are to a significant degree unmappable using short read HTS methods, a point that is often lost in the simplified representations of data used in the relevant studies.

5) The legend or point of Figure 2 A versus B wasn't very clear.....what is the distinction here, whether the model focuses on RepA or not? The Figure 3 concept was clear enough after rechecking the text description (cycles of anchoring versus single anchoring and then chromatin movement) but the figure itself was not easily digested. A model that showed an alternative to "channels", rather than so many modifications of the channel concept, might have been a more helpful option.

I have modified Figure 2 legend to clarify and have simplified Figure 3 (now Figure 4, see below) somewhat. As above reference to channels has now been taken out.

6) The model also doesn't show the chromosome territory, and seems to avoid the question of how Xist is limited to that chromosome? Am I correct that the implication is that the spread would be limited by the amount of Xist?

I have added a new Figure (Figure 3) to illustrate the central concepts of the proposed model for how Xist abundance and anchoring dynamics could dictate range of action from the site of synthesis (associated text on p14). The former Figure 3, now Figure 4, is concerned with models for how Xist RNA translocates from one site to another (diffusion vs alternatives) and this is now linked to the text on p17.

7) If memory serves correctly, I believe the Gilbert paper concluded that the ATP dependent oligomerization of SAF-A was linked to active transcription and open euchromatin, and I believe may have indicated that SAF-A was not on heterochromatin. In any case, is this the opposite of what is proposed here, where polymerization closes rather than opens chromatin?

It is correct that the authors of the Gilbert paper speculate association of SAF-A oligomerisation with chromatin opening linked to active transcription and this is now mentioned (p10). However, it is not straightforward to make parallels between SAF-A action at autosomal sites, where it is proposed to facilitate gene activity, and on Xi where it is proposed to facilitate gene silencing. One factor not considered in the published work is transcription of repeat elements, for example young L1 LINE elements which are transcribed on Xi during establishment of X inactivation.

Referee: 2

Comments to the Author(s)

The review "Localised accumulation of Xist RNA in X chromosome inactivation" describes recent findings in X chromosome inactivation. In particular, the following important question is considered in details: how Xist RNA accumulates locally on the chromosome from which it is transcribed. The author suggest several models of localised accumulation of Xist RNA in cis accompanied by schemes. The manuscript provides an important contribution to the field, is well-written and comprehensive.

However some questions about the described mechanisms of Xist RNA retention in the chromatin of inactive X chromosome arise:

1) It is stated that "Xist RNA is anchored through interactions with proteins of the nuclear matrix". However, interaction of Xist RNA with SAF-A/hnRNPU can be also explained by co-transcriptional binding of this protein to nascent transcript and formation of a stable RNP particle. It is also known

that SAF-A/hnRNPU could have a prion-like activity and thus form a biomolecular condensate leading to formation of dynamic membraneless compartments. Thus it is better to use this modern terminology instead of rigid "nuclear matrix".

There is presently no evidence to support a role for membraneless compartments linked to Xist bound proteins, including SAF-A, in X inactivation and Xist behaviour. Whilst this represents an interesting and topical idea, exploring the idea in any depth is beyond the scope of this review and probably premature.

2) Chromosome conformation capture approaches revealed global loss of local structure on the inactive X chromosome and formation of large mega-domains. I suggest to discuss the possibility of preferential spreading of Xist RNA along inactive X chromosome due to the absence of such local structure including active/inactive compartments and topologically associating domains with their insulating properties.

Again, an interesting and topical issue. I do draw attention to how 3D chromatin contact frequency is a potential driver of preferred Xist binding sites during early establishment of X inactivation and I allude to the possible importance of compartments in limiting Xist accumulation to gene rich domains (p13). More recent studies indicate that restoring TADs on Xi in somatic cells does not affect either gene silencing or local accumulation of Xist RNA, suggesting that topology is less important once X inactivation is fully established.